# Prediction of Temperature and Viscosity Profiles in Heavy-Oil Producer Wells Implementing a Downhole Induction Heater

**Javier Ramírez [1,\*], Alexander Zambrano [2] and Nicolás Ratkovich [1]**

[1] Department of Chemical and Food Engineering, Universidad de los Andes, Cra. 1 N °18A—12, Bogotá 111711, Colombia
[2] BCPGroup Artificial Lift, Autopista Medellín Km 0+440 Mts., Tenjo 250208, Colombia
[\*] Correspondence: ja.ramirezs2@uniandes.edu.co; Tel.: +57-(350)-345-2211

**Abstract:** Very high viscosity significantly impacts the mobility of heavy crude oil representing difficulties in production and a decrease in the well's efficiency. Downhole electric heating delivers a uniform injection of heat to the fluid and reservoir, resulting in a substantial decrease in dynamic viscosity due to its exponential relationship with temperature and a drop in frictional losses between the production zone and the pump intake. Therefore, this study predicts temperature and viscosity profiles in heavy oil-production wells implementing a downhole induction heater employing a simplified CFD model. For the development of the research, the geometry model was generated in CAD software based on the geometry provided by the BCPGroup and simulated in specialized CFD software. The model confirmed a 46.1% effective decrease of mean 12°API heavy-oil dynamic viscosity compared with simulation results without heating. The developed model was validated with experimental data provided by the BCPGroup, obtaining an excellent agreement with 0.8% and 15.69% mean error percentages for temperature and viscosity, respectively. Furthermore, CFD results confirmed that downhole electrical induction heating is an effective method for reducing heavy-oil dynamic viscosity; however, thermal effects in the reservoir due to heat penetration were insignificant. For this study, the well will remain stimulated.

**Keywords:** enhanced oil recovery (EOR); reservoir and well performance; downhole induction heater; heavy-oil producer wells; CFD

## 1. Introduction

Heavy crude oil is defined as a high-density, highly viscous liquid petroleum. The generic term heavy oil is often applied inconsistently to crude oil that has an API gravity of less than 20°, other definitions classify heavy oil as an oil having an API gravity less than 22° API, or less than 25° API [1]. Extraction of this type of crude oil is much more difficult and less efficient compared to conventional crude oil [2]. Most of the world's oil resources, especially the ones produced in South and North America are viscous and heavy hydrocarbons [3]. These reservoirs have specific characteristics that make oil extraction a complex process, due to high viscosity, which generates resistance to flow through the integrated production system [4] making it necessary to develop techniques to facilitate their extraction such as thermal enhanced oil recovery. According to Elam et al. [5] thermal conductivity for the crude oils is rather insensitive to changes in temperature for temperatures from 273 to 323 K.

Tarom and Hossain [6] developed a semi-analytical method for temperature profile prediction in injection/production wells. Their approach was to predict temperature as a function of depth for injection production wells using the developed model. Numerical simulation with ANSYS Fluent [7] (CFD Software, STAR-CCM+ v16.06.010-R8) was used to validate the results. The main result was a significant correlation between the

semi-analytical algorithm and ANSYS Fluent results, demonstrating that both the semi-analytical model and CFD software are powerful tools to predict temperature profiles along a production/injection wellbore.

Mao et al. [8] carried out a study to quantify accelerated production rates achieved by the installation of wellbore heaters in heavy oil producer wells. Production improvement factor (PIF) was used to evaluate the influence of wellbore heating on three types of oils: medium crude oil, heavy crude oil, and extra heavy oil. It was found that the maximum PIF was approximately 3, a value that is much smaller than that found in published literature, that was in the order of 10–100.

Renaud et al. [9] studied 30 years of production of a deep borehole heat exchanger through numerical simulation and CFD approaches. Results showed that during the first year of production, output temperature was a function of working fluid velocity and that between 1 and 10 years of production cooling perturbation increased radially from 10 to 40 m. Concluding that deepening the internal well, enhances the deep borehole heat exchanger heat transfer, showing 2–3% improvement in comparison with the standard design.

As can be noted from the state-of-the-art previously presented, developed research regarding the study of effective viscosity reduction in different wellbore regions is limited. In the presented research, a general analysis is developed, and global results are presented; however, there is no region-by-region analysis or experimental results comparison. Hence, the main goal of this study is to decompose analysis for important wellbore regions and compare CFD results for temperature and dynamic viscosity with experimental results provided by the BCPGroup.

The thermally enhanced oil recovery technique used in this paper was developed by the BCPGroup, it consists in a system that allows changes in downhole fluid dynamics using thermal methods. These methods efficiently reduce crude-oil viscosity, increasing its mobility and desired fluid characteristics.

The heat generated by the device reduces damage in near wellbore-reservoir areas and can be used with different artificial lift systems. The system is implemented in wells that require Electric Enhanced Oil Recovery (EEOR Projects), achieving an improvement of wellbore productivity by crude-oil viscosity reduction, a mitigation of paraffin and asphaltene precipitation issues and a diminution of system duty caused by heavy crude oil [10].

Key performance indicators (KPI's) proposed by the BCPGroup include: (a) service life increase (30–50%), (b) viscosity recution (20–50%), (c) %BSW reduction (5–15%), (d) energetic consumption reduction (25–60%), (e) operation cost reduction (10–20%) and (f) structural charge reduction (20–30%).

The present research seeks: (a) to validate the viscosity reduction KPI proposed by the BCPGroup, (b) to expand the current state of the art in thermally enhanced oil recovery techniques, (c) to present significant results and conclusions based on CFD temperature and viscosity profiles in downhole regions for heavy crude oil wells. CFD was proven a valuable tool for heavy-crude-oil fluid dynamics and thermal properties prediction due to its high concordance with experimental results.

## 2. Materials and Methods

### 2.1. Geometry Modelling and Mesh Generation

The geometry model was generated in detail using the software Autodesk Inventor Professional 2021 [11] based on the geometry provided by the BCPGroup and the system proposed by Sharma et al. [12].

Wellbore penetration was specified as 9.14 m (30 ft) by BCPGroup; however, to simplify well perforating and crude-oil inlet into the annulus, reservoir, cement, and casing regions were extended. This extension was calculated as three times the diameter of the cross-section, 1.14 m (3.75 ft). Table 1 summarizes axial specification and Figure 1 details visually the wellbore's axial section. Inner and outer diameters for each cross section are specified in

Table 2 and the wellbore's cross-section is detailed visually in Figure 2 . Finally, the heater section specification is detailed in Table 3.

**Table 1.** Axial geometry specifications.

| Section | Length [m] | Length [ft] |
|---|---|---|
| Heater $(l_1)$ | 7.47 | 24.50 |
| Tubing $(l_2)$ | 9.14 | 30.00 |
| Extension $(l_3)$ | 1.14 | 3.75 |
| Total $(l_4)$ | 10.29 | 33.75 |
| Cross Section $(l_5)$ | 0.38 | 1.25 |

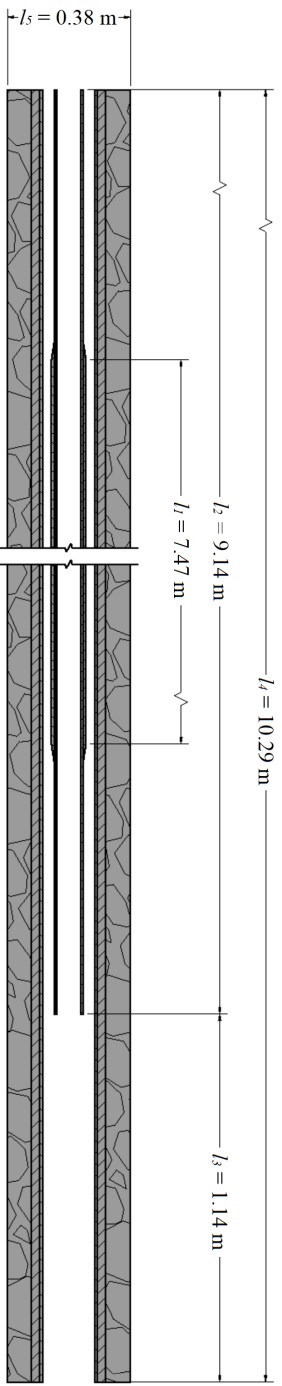

**Figure 1.** Wellbore axial section schematic.

**Table 2.** Radial geometry specifications.

| Region | Material | Inner Diameter [cm] | Outer Diameter [cm] |
|---|---|---|---|
| Tubing | Steel | 7.600 ($d_1$) | 8.890 ($d_2$) |
| Heater | Copper | 8.890 ($d_2$) | 10.668 ($d_3$) |
| Casing | Steel | 15.941 ($d_4$) | 17.780 ($d_5$) |
| Cement | Concrete | 17.780 ($d_5$) | 22.860 ($d_6$) |
| Reservoir | Sandstone | 22.860 ($d_6$) | 38.100 ($d_7$) |

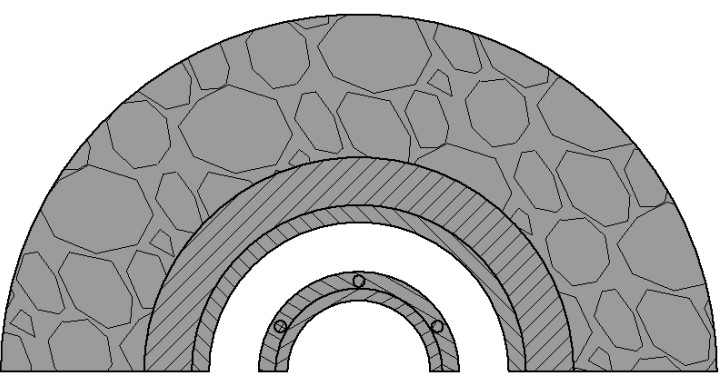

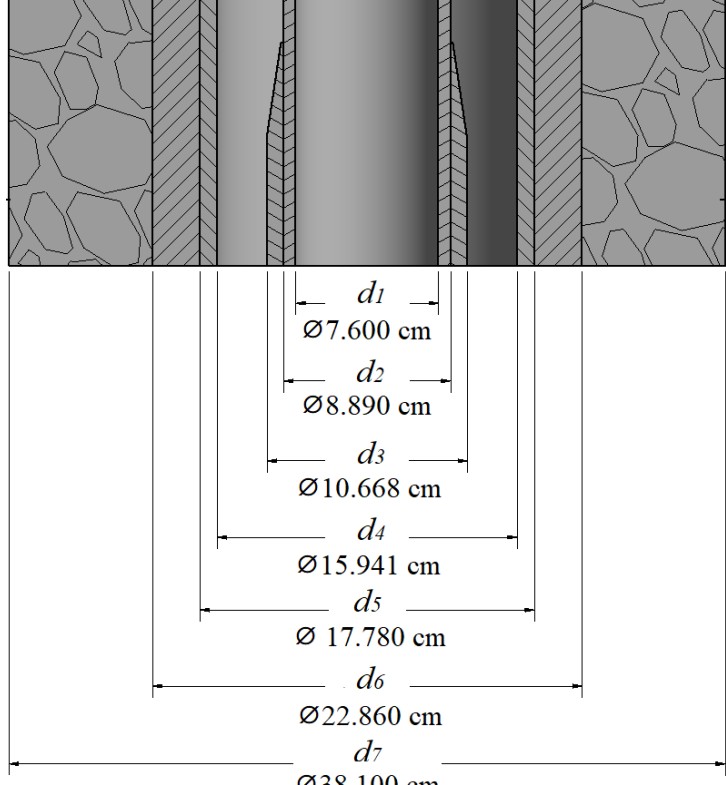

**Figure 2.** Wellbore cross section schematic.

**Table 3.** Heater specifications.

| Parameter | Value |
|---|---|
| Number of electric resistances | 6 |
| Resistance diameter [cm] | 0.635 |
| Resistance length [m] | 7.470 |
| Static temperature [K] | 422.04 |

The internal volume of the CAD model was extracted in CFD software STAR-CCM+ v16.06.010-R8 [13], obtaining the fluid region. Models used for mesh generation include [14]:

- Surface remesher.
- Polyhedral mesher.
- Thin mesher.
- Prism layer mesher.
- Automatic surface repair.

Three prism layers with a stretching factor of 1.5 were located in near-wall regions to properly capture the hydrodynamic layer. Three thin layers were generated for thin geometries, where good quality cells are required to capture the solid material thickness. The online tool Volupe grid calculator [15] was used as a guide for establishing mesh parameters.

To determine the ideal number of cells for the mesh, based on computational cost and accuracy, a mesh independence test (MIT) was carried out [16]. Three different mesh sizes were evaluated (coarse, base, and fine), maintaining other parameters relative to base size.

The top annular temperature was chosen as the variable for mesh comparison, due to the existence of experimental data for this value. For the MIT, the error percentage for top annular temperature was calculated for each mesh. Results for error and computational time are presented in Figure 3.

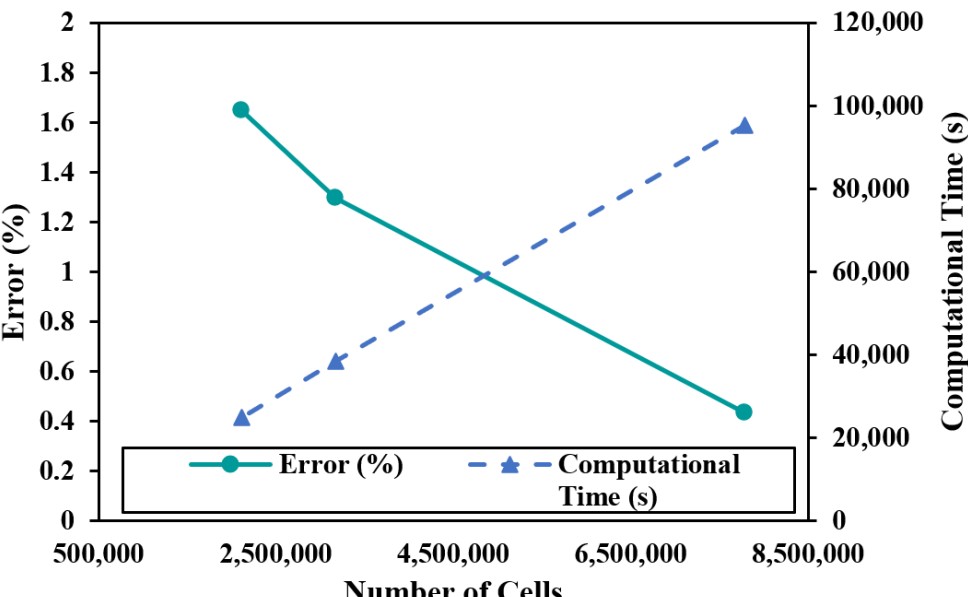

**Figure 3.** Error for top annular temperature [%] and computational time [s] as a function of the number of cells.

As expected, Figure 3 showed that an increase in the number of cells represents an increase in results accuracy. However, the computational time is compromised with an increase in the number of cells. As seen in Figure 3, the most significant error reduction happens in the transition from the base mesh to the fine mesh (0.21%); however, computational time for the fine mesh is almost three times that of the base mesh. Even though error percentages are relatively low for all meshes, the fine mesh, with ~7.8 million cells, represents the best option if adequate computational resources are available, reason why this mesh was selected for the simulations. Finer meshes were carried out; however, no significant variations were observed in comparison with the fine mesh. In the case of limited computational resources, no significant difference was found as of using the coarse or base mesh using top annular temperature as the variable for mesh comparison.

The main parameters for the chosen mesh are detailed in Table 4 and Figure 4 shows the final 3D mesh generated.

**Table 4.** Mesh specification.

| Parameter | Value |
|---|---|
| Base Size [mm] | 7.5 |
| Target Surface Size [mm] | 7.5 |
| Minimum Surface Size [mm] | 3.75 |
| Number of Thin Layers | 3 |
| Number of Prism Layers | 3 |
| Prism Layer Stretching | 1.5 |
| Prism Layer Total Thickness [mm] | 0.75 |

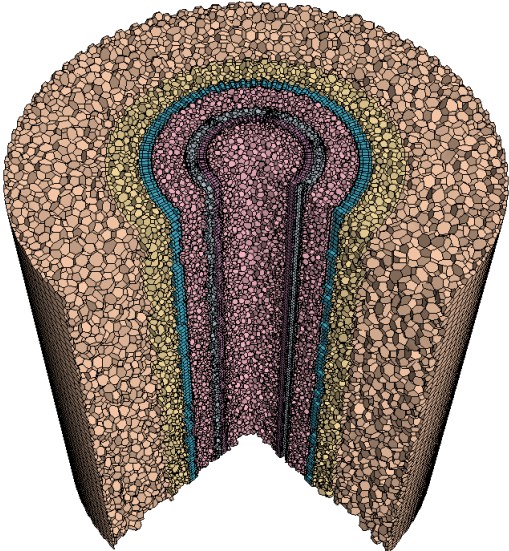

**Figure 4.** Volume mesh generated in STAR-CCM+.

*2.2. Physical Models Specification*

For this study, the system was modeled in a steady state. For fluid, single phase, segregated flow, laminar flow, and constant density were assumed.

Based on experimental oil field viscosity ($\mu_0$), BCPGroup calculated the Reynolds number correlation at several temperatures using Equation (1). Results are shown in Table S1 of supplementary material. A plot for Reynolds number dependent to temperature is shown in Figure 5.

$$N_{Re} = \frac{\rho v D}{\mu_0} \tag{1}$$

Characteristic length for Reynolds number calculation is specified as 15.941 cm (6.276 in), the diameter of the annular section of the wellbore. Velocity ($v$) was calculated with Darcy's Law for flow in a porous medium, as shown in Equation (2), where $q$ represents the Darcy flux, $Q$ the oil's mass flow rate, $A$ the cross sectional area and $\chi$ the porosity of the reservoir [14].

$$v = \frac{q}{\chi} = \frac{Q}{A \cdot \chi} \tag{2}$$

Oil density was modeled as constant with temperature, and it was calculated on the base of oil ° API and through its specific gravity as shown in Equations (3) and (4) [17].

$$\gamma_0 = \frac{141.5}{°API + 131.5} \tag{3}$$

$$\rho = \gamma_0 \rho_{H_2O} \tag{4}$$

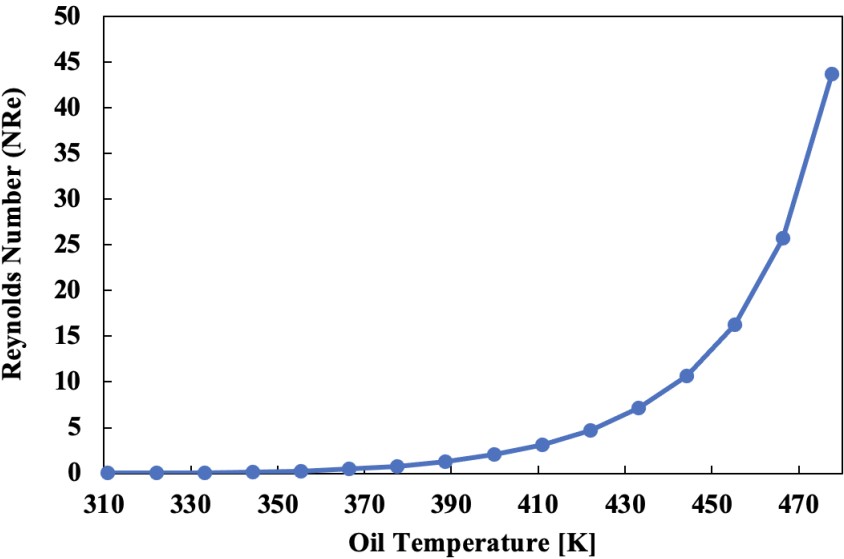

**Figure 5.** Reynolds number and dynamic viscosity function of temperature.

As shown in Figure 5, even if the Reynolds number varies with temperature in the performed analysis, the flow margin does not change, keeping a laminar flow. Due to this, the oil's heat transfer coefficient is considered constant.

Since dynamic viscosity is one of our properties of interest, it was modeled as temperature dependent and specified by a 7th-degree polynomial, obtained by a polynomial curve fitting of experimental data provided by the BCPGroup. MATLAB [18] *polyfit* function with centering and scaling [19] was used to obtain the polynomial coefficients. Equation (5) shows the polynomial obtained where $\mu$ units are [Pa·s] and $T$ units are [K].

$$\mu = -2.1238 \times 10^{-13}T^7 + 5.6677 \times 10^{-10}T^6 - 6.4744 \times 10^{-7}T^5 + 4.1040 \times 10^{-4}T^4$$
$$- 1.5591 \times 10^{-1}T^3 + 3.5500 \times 10^1 T^2 - 4.4860 \times 10^3 T + 2.4271 \times 10^5 \quad (5)$$

Properties for crude-oil and reservoir are presented in Table 5.

**Table 5.** Fluid and reservoir specification.

| Parameter | Value |
|---|---|
| Reservoir temperature [K] | 333.15 |
| Reservoir static pressure [kPa] | 6894.76 |
| Oil gravity [° API] | 12 |
| Production rate [BFPD] | 10 |
| Mass flow rate [kg/s] | 0.105 |
| Porosity | 0.26 |
| Tortuosity | 1.00 |

Regions outer to the annulus (casing, cement, and reservoir) were established as porous regions with porosity and tortuosity defined as in Table 5 to simplify well perforating and crude-oil inlet into the annulus. Solid thermal conductivity was defined for each of these regions. A summary of all regions with their type and relevant properties is presented in Table 6.

Porous regions mathematical modeling is based on porosity ($\chi$), a physical property defined as the ratio of the volume of pores to the volume of bulk rock [20], it may also be defined as the ratio of the volume ($V_f$) that is occupied by the fluid and the total volume ($V$) of a cell [14], as shown in Equation (6). Porous media modeling uses porosity and adds appropiate source terms to the governing equation [14].

**Table 6.** Regions specification.

| Region | Type | Material | Density [kg · m$^{-3}$] | Thermal Conductivity [W · m$^{-1}$· K$^{-1}$] | Specific Heat [J · kg$^{-1}$· K$^{-1}$] |
|---|---|---|---|---|---|
| Fluid | Fluid | Oil | 953 | 0.20 | 2130 |
| Tubing | Solid | Steel | 8000 | 50.20 | 510 |
| Heater | Solid | Copper | 8960 | 400.00 | 385 |
| Heater Casing | Solid | Steel | 8000 | 50.20 | 510 |
| Casing | Porous | Steel | 8000 | 50.20 | 510 |
| Cement | Porous | Concrete | 2242 | 1.73 | 754 |
| Reservoir | Porous | Sandstone | 739 | 2.40 | 835 |

For porous media flow, the momentum equation contains a term that accounts for the resistance to the flow imparted by the porous medium. This term ($f_p$) is defined in terms of superficial velocity ($v_s$) and the porous resistance tensor (P) as shown in Equation (7) [14]. Superficial velocity ($v_s$) is an artificial flow velocity that assumes that neglects the solid portion of the porous medium and accounts for the increase in physical velocity as flow enters a porous media due to the reduction of open area available to the flow. Superficial velocity is related to physical velocity as shown in Equation (8). On the other hand, the porous resistance tensor (P) consists of two components as shown in Equation (9), $P_v$ known as viscous resistance tensor and $P_i$ known as inertial resistance tensor [14].

$$\chi = \frac{V_f}{V} \tag{6}$$

$$f_p = -P \cdot v_s \tag{7}$$

$$v_s = \chi \cdot v \tag{8}$$

$$P = P_v + P_i \mid v_s \mid \tag{9}$$

Pinilla et al. [21] previously calculated the viscous and intertial porous resistances for heavy oil reservoirs using a mathematical regression for the pressure drop against flowrate satisfying Equation (9), known as the Dupuit-Forchheimer equation. These values are presented in Table 7.

**Table 7.** Reservoir's viscous and inertial resistances

| Parameter | Value |
|---|---|
| $P_v$ [kg/m$^3$s] | $7.03 \times 10^7$ |
| $P_i$ [kg/m$^4$] | $1.95 \times 10^8$ |

Two main boundary conditions were established, the first one as an outlet on the corresponding fluid outlet surface, the second one as a mass flow inlet with mass flow rate specified in Table 5 on the corresponding outer surface of the reservoir porous region. The location of these surfaces in the geometry is shown in Figure 6.

Lastly, for heat transfer, a thermal specification is defined for the interface formed by the heating elements (6 lineal resistances) and its casing. The thermal specification in this boundary is defined as temperature, meaning that the boundary temperature as a scalar profile is entered accordingly to the heater static temperature established in Table 3.

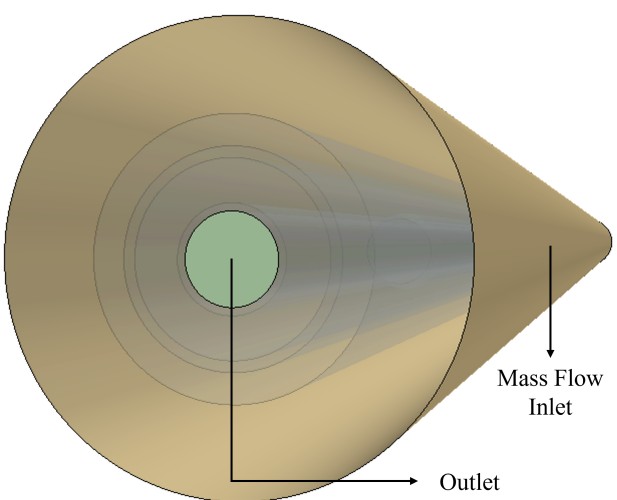

**Figure 6.** Boundary conditions defined in STAR-CCM+ (Green stands for outlet and gold for mass flow inlet).

For heat transfer across the geometry, Fourier's law is applied for heat conduction, as shown in Equation (10) where $\dot{q}''$ [W/m$^2$] is the local heat flux vector, $k$ [W/mK] is the thermal conductivity of the material, and $\nabla T$ [K/m] is the temperature gradient.

$$\dot{q}'' = -k\nabla T \tag{10}$$

Newton's law of cooling governs convective heat transfer at a surface, as shown in Equation (11) where $\dot{q}_s''$ is the local heat flux vector, $h$ is the local convective heat transfer coefficient, $T_s$ is the surface temperature and $Tref$ is a characteristic temperature of the fluid moving over the surface.

$$\dot{q}_s'' = h(T_s - T_{ref}) \tag{11}$$

Finally, for conjugate heat transfer (CHT) between fluid and solid regions at a contact interface, two boundaries are defined: Boundary$_0$ in the fluid region and Boundary$_1$ in the solid region. Based on energy conservation, total heat flux is conserved across the interface, as shown in Equation (12) where $\dot{q}_0$ is the heat flux from the fluid through Boundary$_0$, $\dot{q}_1$ is the heat flux leaving through Boundary$_1$ into the solid and $S_u$ is the heat source within the interface (if applicable).

$$\dot{q}_0 + \dot{q}_1 = -S_u \tag{12}$$

Heat fluxes can be expressed in terms of linearized heat flux coefficients as shown in Equations (13) and (14) for Boundary$_0$ and Boundary$_1$ respectively. Where $A, B, C$ and $D$ are the linearized heat flux coefficients. $T_{c0}, T_{c_1}$ are the cell temperatures next to Boundary$_0$ and Boundary$_1$ respectively and $T_{w0}, T_{w_1}$ are the interface temperatures on the fluid side (Boundary$_0$) and on the solid side (Boundary$_1$) respectively.

$$\dot{q}_0 = A_0 + B_0 T_{c0} + C_0 T_{w0} + D_0 T_{w0}^4 \tag{13}$$

$$\dot{q}_1 = A_1 + B_1 T_{c1} + C_1 T_{w1} + D_1 T_{w1}^4 \tag{14}$$

## 3. Analysis of Results

### 3.1. Velocity

Analysis of velocity profiles is a critical step for understanding the fluid's behavior in different regions such as its outlet, inlet, annulus, and pipe. They are also powerful tools to confirm appropriate flow direction and regime in critical regions. Figures 7 and 8 represent the axial fluid's velocity profile at the tubing outlet and inlet, respectively. Methodology used for the visualization of the vector field is line integral convolution (LIC).

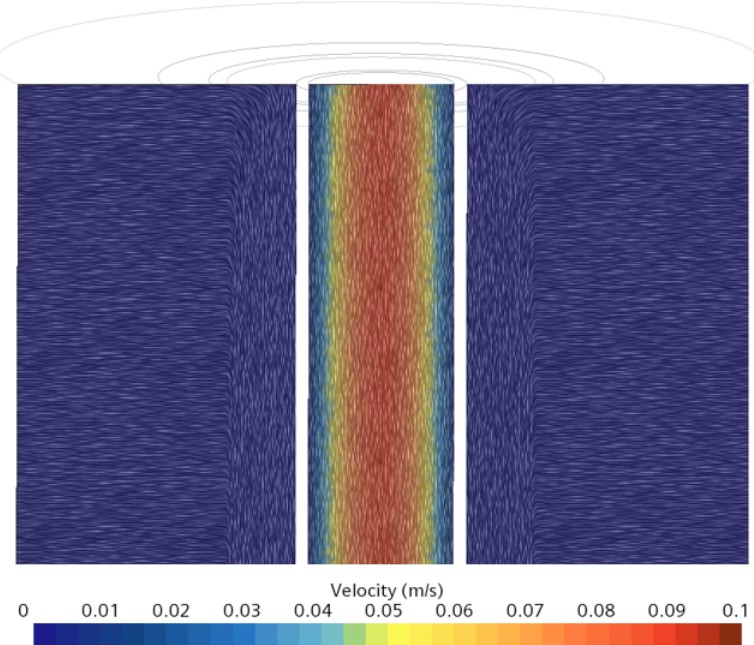

**Figure 7.** Velocity scene for tubing outlet.

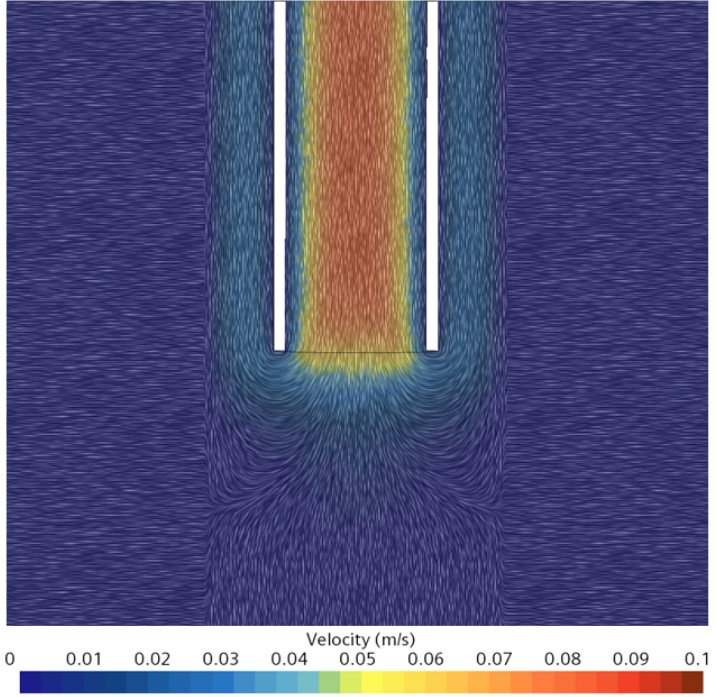

**Figure 8.** Velocity scene for tubing inlet.

Figures 7 and 8 show that velocity magnitude is considerably low as fluid emerges from the reservoir and flows through porous regions, this is due to the large surface area specified as fluid inlet and resistance exerted by the porous region. On the other hand, as fluid enters the annulus, its velocity increases and reaches values near $0.05 \frac{m}{s}$ on tubing entry. As fluid flows through the pipe, its maximum velocity is achieved, with a value of nearly $0.10 \frac{m}{s}$. It can also be seen that the fluid moves without lateral mixing, and no cross flows, eddies, or swirling motion.

### 3.2. Temperature and Viscosity

Obtaining and analyzing temperature and viscosity profiles for the fluid is one of the main objectives of this study. To analyze the fluid's total temperature and dynamic viscosity from both a radial and axial perspective, four critical axial points were identified: (a) tubing inlet, (b) heater start point, (c) heater end point, and (d) fluid outlet.

Radial planes were created in each of these regions and temperature and viscosity scenes were obtained for the fluid region, as shown in Figures 9 and 10.

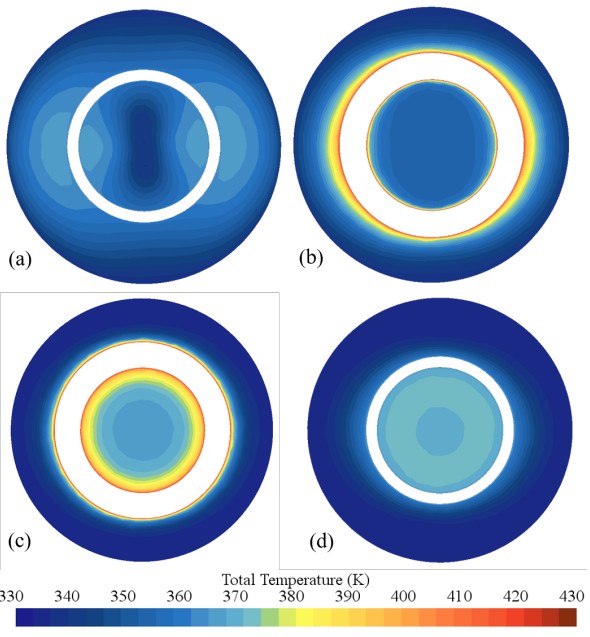

**Figure 9.** Total temperature profile for tubing inlet (**a**), heater start (**b**), heater end (**c**) and outlet (**d**).

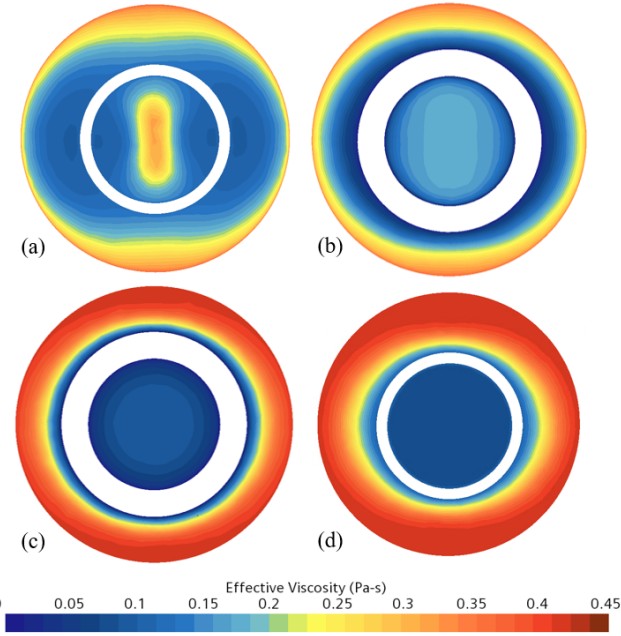

**Figure 10.** Effective viscosity profile for tubing inlet (**a**), heater start (**b**), heater end (**c**) and outlet (**d**).

It should be noted that viscosity is strongly dependent on temperature, with inverse proportionality, which means that at lower temperatures a higher viscosity is expected and low viscosities are associated with hot regions [22].

Figures 9a and 10a show temperature and viscosity profiles at the tubing inlet. It can be seen that lower temperatures and high viscosities concentrate on the pipe's center, due principally to flow coming from the lower region of the annulus, where temperatures are lower due to its distance to the heater. On the other hand, the annular region near the pipe has a higher temperature and lower viscosity, explained by the fact that most of this flow, comes from upper annular regions where fluid temperature is highly influenced by the heater's heat flow.

The start of the heating region is another critical point for the fluid's temperature and therefore viscosity. Figures 9b and 10b illustrate temperature and viscosity at this point. As expected, heat transfer has caused a temperature increase and viscosity decrease in the pipe's fluid in comparison with the tubing inlet. For annular fluid, the radial temperature average is at its highest point, because the annular upper flow has been heated along all the 7.47 m heater length and has reached the end of the heating section.

Temperature and viscosity profiles at the end of the heating section (from the pipe flow perspective) are shown in Figures 9c and 10c. At this point tubing fluid temperature has reached its maximum temperature and lowest viscosity since it has been heated along the heating section. On the other hand, annular fluid has a decreased temperature and increased viscosity in comparison with the heater's start region, this is explained by the fact that this flow comes mainly from the system's mass flow inlet, and no heat has been applied to this fluid since the annulus flow direction is downwards.

Lastly, the system's outlet is considered the fluid's last relevant point. At this stage, the pipe's fluid temperature has stabilized radially due to the absence of a heat source, reaching a final temperature of 370.65 K and an average viscosity of 0.106 Pa-s. Annular fluid is mainly at its inlet temperature of 333.15 K and initial viscosity of 0.423 Pa-s; however, a slight change of these properties is observed at the near-pipe region due to heat transfer from the pipe's fluid.

## 4. Discussion of Results

Experimental data provided by the BCPGroup consists of fluid viscosity and temperature measured in the annular up and down regions. CFD results at these measuring points were estimated as STAR-CCM+ surface averages [14] for the previously specified areas. Table 8 presents experimental and CFD data and calculated errors for both temperature and viscosity.

**Table 8.** Experimental and CFD comparison.

|  |  | Annular Up | Annular Down |
|---|---|---|---|
| Temperature [K] | Experimental | 339.82 | 328.26 |
|  | CFD | 339.43 | 333.15 |
|  | Error [%] | 0.11 | 1.49 |
|  | Total Error [%] | 0.8 | |
| Viscosity [Pa-s] | Experimental | 0.3179 | 0.5547 |
|  | CFD | 0.3420 | 0.4227 |
|  | Error [%] | 7.58 | 23.80 |
|  | Total Error [%] | 15.69 | |

As shown in Table 8, the results for the upper annular region were more accurate than the ones for the lower annular region. This is explained mainly by the 1.14 m (3.75 ft) geometrical extension of the annular region detailed in Table 1 and Figure 1. This extension affects flow behavior and properties. On the other hand, temperature results were found to be more accurate than viscosity ones, this behavior was expected, because the polynomial regression for viscosity, detailed in Equation (5) was obtained by a polynomial fitting of experimental data, and a considerable error source exists in this procedure. It is important to note that viscosity has a sensitive behavior to any change in temperature, so correlations between these two variables usually induce significant sources of deviation.

To calculate CFD effective viscosity reduction, a volume average of dynamic viscosity was calculated by STAR-CCM+ in the fluid region with the downhole heater, obtaining a viscosity of 0.228 Pa-s, this value was compared to dynamic viscosity at reservoir temperature, neglecting fluid thermal variations, obtaining 0.423Pa-s and an effective viscosity reduction of 46.1%.

Finally, it is important to highlight that thermal effects on regions further than the annulus fluid were not analyzed in this study, since that CFD results indicated that heat transfer further than the annulus region was not significant.

## 5. Conclusions

- A numerical model based on CFD codes was carried out to simulate the performance of an induction downhole heater with a constant temperature of 422.04K in a heavy oil well. Temperature and viscosity profiles were presented for relevant regions, and results were validated against experimental data from the BCPGroup, obtaining mean error percentages of 0.8% and 15.69% for temperature and viscosity, respectively.
- Obtained results validated the key performance indicator (KPI) referring to viscosity reduction proposed by the BCPGroup and estimated between 30% and 50%. CFD showed an effective viscosity reduction of 46.1% compared to the same system without the heater (neglecting fluid thermal variations).
- An excellent agreement with experimental data was found for annular up and annular down temperatures, obtaining error percentages of 0.11% and 1.49%, respectively.
- For viscosity results, error percentages of 7.58% and 23.80% were obtained for annular up and annular down temperatures, respectively. This behavior is explained by the error source introduced by the polynomial regression for viscosity, since viscosity is highly sensitive to changes in temperature, especially in heavy oils.
- This study confirmed the benefits and necessity of the implementation of a thermally enhanced oil recovery technique for heavy-crude-oil wells since its impact on viscosity is highly positive and improves heavily well performance.
- Further studies are necessary to confirm other KPI's proposed by BCPGroup, such as service life increase, energetic consumption reduction, and operation cost reduction. However, it is highly predictable that results will confirm the benefits of the referred technique.
- A better heat penetration in near-well regions is expected in further electric adaptations to the heater achieving an increase in the wellbore's productivity. In future studies with wellbores deeper than 3658 m (12000 ft), an electrical and thermal analysis should be carried out between the drillings and well surface to maintain an adequate wellhead viscosity for fluid transportation to processing stations without major inconveniences.

**Supplementary Materials:** The following supporting information can be downloaded at: https://www.mdpi.com/article/10.3390/pr11020631/s1, Table S1: Dynamic Viscosity and Reynolds Number of Oil function of temperature.

**Author Contributions:** Conceptualization, A.Z. and N.R.; Data curation, J.R.; Formal analysis, J.R.; Funding acquisition, N.R.; Investigation, J.R.; Methodology, J.R. and N.R.; Project administration, N.R.; Resources, A.Z. and N.R.; Software, J.R. and N.R.; Supervision, A.Z. and N.R.; Validation, A.Z.; Visualization, J.R.; Writing—original draft, J.R.; Writing—review and editing, A.Z. and N.R. All authors have read and agreed to the published version of the manuscript.

**Funding:** This research received no external funding.

**Institutional Review Board Statement:** Not applicable.

**Informed Consent Statement:** Not applicable.

**Data Availability Statement:** Data is contained within the article or supplementary material.

**Acknowledgments:** The authors gratefully acknowledge the Information and Technology Department of the University of Los Andes (DSIT) for supplying the hardware to conduct this research.

**Conflicts of Interest:** The authors declare no conflict of interest.

**Abbreviations**

The following abbreviations are used in this manuscript:

| | |
|---|---|
| CFD | Computational Fluid Dynamics |
| EOR | Enhanced Oil Recovery |
| API | American Petroleum Institute |
| PIF | Production Improvement Factor |
| EOR | Enhanced Oil Recovery |
| EEOR | Electric Enhanced Oil Recovery |
| BSW | Basic Sediment and Water |
| KPI | Key Performance Indicator |
| CAD | Computer Aided Design |
| MIT | Mesh Independency Test |
| CHT | Conjugate Heat Transfer |
| LIC | Line Integral Convolution |

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
