# Peer review of "Prediction of Temperature and Viscosity Profiles in Heavy-Oil Producer Wells Implementing a Downhole Induction Heater"

_processes, doi:10.3390/pr11020631_

Round 1
Reviewer 1 Report
Generally, the considered topic is interesting, but not new. I have several comments that need to be clarified before publication. My major concern is about the considered model statement description, which suffer from the missing some important information. Such, it is not clear how the porous regions were exactly modelled. Then, what type of boundary condition was applied to the heater surface and why. In the text was mentioned, that “The thermal specification in this boundary is defined as temperature, meaning that the boundary temperature as a scalar profile is entered accordingly to the reservoir temperature established in Table 5.” If constant temperature is used at this boundary it would mean that the heater is generally would have unlimited heat source at these conditions.
What dimension is used as characteristic length in the Re number. Regarding the results, how authors could explain the unsymmetrical temperature profile at the tubing inlet for example (Figure 9a) assuming that the problem formulation is fully symmetrical.
As per journal requirements, SI units should be used.
Reviewer 2 Report
CFD simulation is carried out for a heavy-crude-oil will in order to analyze the influence mainly in temperature and viscosity of an induction wellbore heater with a constant temperature. The results were analyzed from a radial and axial perspective, taking into consideration fluid outlet and fluid properties. These results are validated with experimental data. From this paper, it is clear that appear necessity of the implementation of a thermal-enhanced oil recovery technique for heavy-crude-oil wells. CFD results confirmed that downhole electrical induction heating is an effective procedure for reducing heavy-oil dynamic viscosity.
Reviewer 3 Report
Please correct your manuscript and submit it.

Reviewer 4 Report
The report presents the outcome of a study on the influence mainly in temperature and viscosity of an induction wellbore heater with a constant temperature of 300°F. The contribution of the report to the body of knowledge is significant and novel. Also, the aim and objectives of the study are within the scope of Processes. The author should consider the following points:
Q1. There are two citations in the abstract. That is not good enough. There is need for the author to structure the abstract. Consequently,
(1) Use one sentence to present the significance of the study, let us know the important of the study.
(2) Use one sentence to present the aim of the study.
(3) Use one sentence to present the research methodology, and
(4) Use two sentences to present the major conclusion drawn from the study.
Q2. It is not really proper to itemize facts under the introduction. Line 62 – 72, merge all the items to form a paragraph on the subject matter.
Q3. Line 73 – 75, it was written, “ The results obtained in this work will validate the viscosity reduction KPI proposed by the BCPGroup and expand the current state of the art in thermally enhanced oil recovery techniques. It will also present significant results and conclusions based on CFD temperature and viscosity profiles in downhole regions for heavy crude oil wells.”
Comment: There is need to revise these sentences. Do not use the word, “will”. Can you even delete? That would make the report looks better. However, present these objectives as a research questions.
Q4. Line 152, it was written, “3. Results”. Change to 3. Analysis of Results
Q5. It is not really proper to itemize facts under the introduction. Line 169 – 172, merge all the items to form a sentence on the subject matter.
Q6. Line 204, it was written, “4. Discussion”. Change to 4. Discussion of Results
Q7. The conclusion section needs revision.
Comment: Start the conclusion section with a sentence stating the aim of the study elaborately. Then remark that the major objectives had been established. Start the conclusion section with a fact on the achievement of the research aim before stating conclusive statements. Revise the title to present the aim concisely. The aim of the title should reflect in the abstract, and at the beginning of the conclusion. Author's should revise the conclusion section to provide conclusive statements on the research questions posed at the end of the introduction. The conclusion section should be revised. Try to itemize all the conclusive facts.
Q8. After the Reference, provide the data used to plot Figure 5 title, “Reynolds number function of temperature.” The data would be more useful by the readers after publication.
Q9. The title was written as, “CFD Modelling of Downhole Induction Heater for Prediction of Temperature and Viscosity Profiles in Heavy-oil Producer Wells” It seems necessary to revise the title. Note that the methodology is CFD Modelling and not really a keyword. The author may consider:
Prediction of Temperature and Viscosity Profiles in Heavy-oil Producer Wells using Downhole Induction Heater
Q10. Update the first paragraph of the introduction with an important fact like, “For temperatures between 273 and 323 K, crude oil's thermal conductivity is relatively unaffected by temperature changes; according to Author 1 et a;. [1].”
Author 1, Author 2, Author 3, Author 4, Author 5, and Author 6 (2022). Ratio of Momentum Diffusivity to Thermal Diffusivity: Introduction, Meta-analysis, and Scrutinization. Chapman and Hall/CRC. New York. ISBN-13: 978-1032108520, ISBN-10: 1032108525, ISBN9781003217374. https://doi.org/10.1201/9781003217374
Round 2
Reviewer 1 Report
I thank the authors for their effort in improving the manuscript. Now clarity has been improved.
Reviewer 3 Report
I see authors answers and thanks. If they also included their corrections to their manuscript that's very good than in that case I can accept manuscript for publication.